# Development of Biodegradable Flame-Retardant Bamboo Charcoal Composites, Part II: Thermal Degradation, Gas Phase, and Elemental Analyses

**DOI:** 10.3390/polym12102238

**Published:** 2020-09-28

**Authors:** Shanshan Wang, Liang Zhang, Kate Semple, Min Zhang, Wenbiao Zhang, Chunping Dai

**Affiliations:** 1Department of Engineering, Zhejiang Provincial Collaborative Innovation Center for Bamboo Resources and High-Efficiency Utilization, Zhejiang A&F University, Hangzhou 311300, China; wangshanshan430@163.com (S.W.); chaselz520103@163.com (L.Z.); zhang888@rish.kyoto-u.ac.jp (M.Z.); 2Department of Wood Science, Faculty of Forestry, University of British Columbia, 2900-2424 Main Mall, Vancouver, BC V6T 1Z4, Canada; katherine.semple@ubc.ca

**Keywords:** bamboo charcoal, aluminum hypophosphite, polylactic acid, composites, flame retardancy

## Abstract

Bamboo charcoal (BC) and aluminum hypophosphite (AHP) singly and in combination were investigated as flame-retardant fillers for polylactic acid (PLA). A set of BC/PLA/AHP composites were prepared by melt-blending and tested for thermal and flame-retardancy properties in Part I. Here, in Part II, the results for differential scanning calorimetry (DSC), thermogravimetric analysis (TGA), Fourier transform infrared (FTIR), thermogravimetry-Fourier transform infrared spectrometry (TG-FTIR), X-ray diffraction (XRD), and X-ray photoelectron analysis (XPS) are presented. The fillers either singly or together promoted earlier initial thermal degradation of the surface of BC/PLA/AHP composites, with a carbon residue rate up to 40.3%, providing a protective layer of char. Additionally, BC promotes heterogeneous nucleation of PLA, while AHP improves the mechanical properties and machinability. Gaseous combustion products CO, aromatic compounds, and carbonyl groups were significantly suppressed in only the BC-PLA composite, but not pure PLA or the BC/PLA/AHP system. The flame-retardant effects of AHP and BC-AHP co-addition combine effective gas-phase and condensed-phase surface phenomena that provide a heat and oxygen barrier, protecting the inner matrix. While it generated much CO_2_ and smoke during combustion, it is not yet clear whether BC addition on its own contributes any significant gas phase protection for PLA.

## 1. Introduction

Polylactic acid (PLA) is a renewable, easily degradable plant starch-based polymer bioplastic that is commonly used in applications such as drug delivery [1], disposable packaging [2], and 3D printing [3]. However, low thermal and moisture stability, as well as flame resistance, reduce the safety and performance of PLA in many applications, including large volume commodity packaging. The focus of this research is to study the combustion kinetics of PLA polymer fortified with naturally sourced bamboo charcoal (BC) and aluminum hypophosphate (AHP), a common mineral fire-retardant filler for polymers such as PLA [4]. BC is very porous, with a high surface area [5], is generally compatible with hydrophobic plastic polymers including PLA, and is expected to improve the fire resistance properties without reducing its mechanical properties based on previous findings by Lau (2014) and Ho et al., (2015) [6,7].

In the first part of this publication series, Wang et al., (2020) [8] gave the composite fabrication processing and mechanical properties/fire retardant tests of PLA with BC and AHP, showing how adding increasing quantities of BC on its own to PLA increased the brittleness and drastically reduced the flexural strength of the material. However, it was found that the co-addition of AHP up to 25% to a base mix of 75% PLA and 25% BC improved the mechanical properties of the composites. The optimum mix for both mechanical properties and flame resistance properties was 25% BC, 25% AHP, and 50% PLA. Fire resistance tests—UL-94 vertical flame test and limiting oxygen index (LOI), and the cone calorimeter test (CCT)—demonstrated how the combined BC + AHP addition drastically reduced (by 72%) the heat release rate and total heat release of the composites during CCT compared with pure PLA. Flame-retardant performance was significantly improved, with the limiting oxygen index (LOI) increased to 31 vol%, providing a V-0 rating in the UL-94 vertical flame test. Early formation of char over the BC/PLA/AHP composites effectively reduced the heat release indices during the cone calorimeter test, improved the fire performance index (FPI), and reduced the fire growth index (FGI). The optimal mix was BC/PLA/AHP 25/50/25 wt. %. The reduced combustibility of the material resulted from the combined effect of halving the quantity of flammable polymer, and the combustion-retarding effects of the fillers. There is a possible, but unconfirmed synergistic effect between BC and AHP, which enhances the flame retardancy of polymer composites and may help reduce the loading of AHP required.

The thermal degradation kinetics of the BC/PLA system has been studied previously by Qian et al., (2016) [9], and that of the AHP/PLA system by Gu et al., (2019) [4]. Liu et al., (2016) [10] studied the flame-retardant properties of a three-phase system involving PLA, AHP, and powdered siliceous rice husk (SRH); in what is perhaps the closest known facsimile to the three-phase system of PLA/BC/AHP tested here. They substituted 5% of the original 20% AHP addition with SRH. The carbon residue of PLA/15AHP/5RHS obtained at 700 °C increased from 12.8% for PLA-AHP to 17.3%, and the onset of thermal degradation of the composite was brought forward. It appears that the addition of a small amount of RHS with its higher thermal stability had a beneficial effect of reducing AHP dosage and increasing the carbon residue, but to a limited extent. We aim to improve the dual action effect by adding a different and possibly more effective filler, BC, in greater quantities as a bulking agent, flame retardant, and halving the quantity of flammable PLA in the composite.

The objective of this work was to gain a better understanding of the combustion kinetics of the combined PLA + BC + AHP system through constituent and residue analyses via differential scanning calorimetry (DSC), thermogravimetric analysis (TGA), Fourier transform infrared (FTIR), thermogravimetry-Fourier transform infrared spectrometry (TG-FTIR), X-ray diffraction (XRD), and X-ray photoelectron analysis (XPS). The possible mechanisms of combustion retardation by the BC and AHP fillers are discussed as the BC and AHP may act synergistically to promote the early heterogeneous nucleation of PLA, forming a dense surface char layer (condensed phase) with a supplementary gas phase (oxygen dilution and consumption) flame-retardant action from the AHP.

## 2. Materials and Methods

The full sequence of composite manufacture and combustion testing was given in Part I. The elemental and thermal properties analysis techniques for the data in Part II are given below.

Differential scanning calorimetry (DSC) tests were conducted with a DSC-500B (Shanghai YANJIN Scientific Instrument Co., Ltd., Shanghai, China) under nitrogen at a flow rate of 50 mL/min. Samples of 10 mg were weighed and sealed in a scaled aluminum pan with an empty pan used as the reference. They were heated from the room temperature to 205 °C at a rate of 10 °C/min^−1^, kept for 5 min, and then cooled down to 0 °C at a rate of 10 °C/min. The samples were heated again to 180 °C at a rate of 10 °C/min. Crystallinity ($X_{C}$) was estimated according to the following Equation (1):(1)$$X_{C}=\frac{∆H_{c}}{∆H_{0}\cdot X_{PLA}}\times 100\%$$
where $∆H_{c}$ refers to the crystallization enthalpy of BC/PLA and BC/PLA/AHP; $∆H_{0}$ refers to the enthalpy value during 100% crystallization of PLA, which is 93.6 J/g; and $X_{PLA}$ refers to the weight ratio of PLA in BC/PLA and BC/PLA/AHP.

Thermal stability (TGA) was determined using a thermo-gravimetric analyzer (F1, NETZSCH Co., Ltd., Selbu, Germany) under a nitrogen atmosphere for BC/PLA and BC/PLA/AHP composites. Approximately 10 mg of sample was placed in a standard ceramic crucible and heated from 50 to 800 °C at a heating rate of 10 °C/min under a nitrogen flow of 20 mL/min. An empty crucible was used as a reference.

Elemental analyses were carried out on the fabricated composites before CCT using an elemental analyzer (Vario EL III, Elementary Co., Ltd., Langenselbold, Germany) in an oxidation furnace at 1200 °C for 10 min. Then, 0.2 ± 0.05 mg of BC was dried and placed in aluminum foil as a control. XPS/XRD analyses were carried out in conjunction with scanning electron microscope (SEM) imagery.

X-ray photoelectron spectroscopy (XPS) was used to analyze the element contents of the residue after CCT. Measurements were performed using a XSAM 800 (KRATOS Co., Ltd., Manchester, UK) spectrometer, with Al kα excitation radiation (*h*_v_ = 1253.6 eV) in ultrahigh vacuum conditions (8 × 10^−^^8^ Pa). 

X-ray diffraction (XRD) profiles were obtained using an X-ray diffractometer (XRD6000, Shimadzu Co., Ltd., Kyoto, Japan). The patterns were collected by Cu Kα (0.15406 nm) radiation. All samples of 5–8 mg were prepared and analyzed under tube voltage of 40 kV, tube current of 30 mA, and scanning range of 2°–80° (2θ) at a rate of 2°/min.

Thermogravimetric analysis-Fourier transform infrared spectrometry (TG-FTIR) was used to elucidate the chemical structure and pyrolysis properties of the constituents and composites. An FTIR analyzer (Prestige-21, Shimadzu Co., Ltd., Kyoto, Japan) was used with a scanning range of 4000–400 cm^−1^. BC was first ground to 200 mesh (75 μm) standard sieve and oven dried. The spectra were obtained using the KBr pellet technique. Pure BC, PLA, and the experimental set of BC/PLA/AHP composites were ground into a powder and then mixed and compressed with KBr powder (about 500 mg) into 13 × 13 × 1 mm^3^ thin discs. A TG Analyzer (TG 209 F3, NETZSCH Co., Ltd, Selbu, Germany) was coupled to a FTIR spectrophotometer (Tensor 27, BRUKER AXS GmbH, Karlsruhe, Germany). All samples were heated from 50 to 800 °C at a heating rate of 10 °C/min in nitrogen atmosphere with flow rate of 70 mL/min. The FTIR spectrometer range was set to 4000 to 400 cm^−1^.

## 3. Results and Discussion

### 3.1. Crystallization Proprties

Differential scanning calorimetry (DSC) was used to explore the thermodynamic properties of composites, that is, glass transition temperature (*T*_g_), cold crystallization temperature (*T*_cc_), crystallization enthalpy ($∆H_{c}$), melting temperature (*T*_m_), and melting enthalpy (Δ*H*_m_). The crystallinity of the material was obtained by calculation. The heat flow curves of PLA, BC/PLA, and BC/PLA/AHP are shown in Figure 1 and the corresponding data are summarized in Table 1.

The *T*_g_ values increased when BC and PLA were added (Table 1). $X_{C}$ values (Table 1) for BC/PLA and BC/PLA/AHP mixes increased (Table 1), showing that BC dispersed uniformly in the polymer matrix plays an important role in heterogeneous nucleation. BC particles provide nucleation sites whereby PLA molecular chains form first on the surface. $X_{C}$ of the 25/75 BC/PLA mix was 2.5 times of that of pure PLA. Adding BC to PLA reduces the cold crystallization temperature (*T*_cc_), that is, it promotes the earlier crystallization of PLA. Double melt crystallization peaks (Figure 1a) appeared with the addition of BC, possibly caused by partial incomplete cold crystallization at a lower temperature, corresponding to the first peak, with the second forming through melting and recrystallization. With the increasing BC content, the melting temperature of the polymer gradually decreased, in keeping with the reduced volume and thickness of the crystalline polymer interphase [11]. Above 35 wt. % BC, the consolidation of the mix decreased sharply, resulting in a non-value for *T*_g_. Although the increase of crystallinity could enhance the rigidity and hardness of the material [11,12], it has a negative effect on the flexural strength and ductility, as shown in Part I, where excessive BC and AHP content >25 wt. % caused problems with agglomeration in the PLA matrix, forming weak interfaces and affecting stress transfer.

The addition of AHP increased the *T*_g_ and *T*_m_ of BC/PLA/AHP mix, but had little effect on the melting temperature. From Table 1, with 15% AHP addition to the 25% BC/PLA mix, the *T*_g_ was 59 (± 1) °C, which was 3 (± 1) °C higher than BC/PLA (25/75). The *T*_m_ was 149 (± 1) °C, or 3 (± 1) °C lower than that of BC/PLA (25/75). AHP caused the double crystal melting peak of the composite to disappear (Figure 1b), indicating that the decrease in crystallinity delays the crystallization process at a certain heating rate. With the increasing AHP content, the *T*_cc_ of BC/PLA increased, but all mixes were stable at 127 (± 2) °C. $X_{C}$ (4.07%) of BC/PLA/AHP (25/50/25) was 85% lower than that of BC/PLA (25/75). The crystallinity of the BC/PLA/AHP mix decreases first and then increases with the increase of AHP content, which is consistent with the findings that the mechanical properties decrease when the content of AHP exceeds 20 wt. %.

### 3.2. Thermal Degradation Properties of Constituent Materials and Composite Mixes

The thermogravimetric (TG) and differential thermogravimetric (DTG) curves for the additives, BC, AHP, and BC/AHP mix under N_2_, are shown in Figure 2a,b, and the corresponding data are listed in Table 2. The initial decomposition temperature can be considered as the temperature at which the weight loss is about 5% [13]. The relative thermal stability of the samples is compared by the temperature of 5% weight loss (*T*_−5%_), the temperature of maximum rate of weight loss (*T*_max_), and the percentage char yield at 600 °C [13]. *R*_peak_ is defined as the maximum weight loss rate, while the *T*_max_ is defined as the temperature at which the samples experience peak mass loss rate (pMLR). Under nitrogen atmosphere, the *T*_−5%_ of AHP was 320 °C. From Figure 2a, the thermal degradation process of AHP was in two stages, *T*_max1_ = 331 °C and *T*_max2_ = 433 °C, respectively; *R*_peak_ was 7.5%/min and 1.6%/min, respectively. AHP decomposed completely at about 759 °C and the residue yield at 800 °C was 65.5%, compared with a 70.8% residue at 800 °C.

Being an already carbonized material, BC underwent no significant weight loss during the pyrolysis process. Only one thermal decomposition peak appears at 649 °C, corresponding to the *R*_peak_ of 0.2%/min, and the final carbon residue rate was 91.4%. Compared with AHP alone, the *T*_−5%_ of the flame-retardant system of compounded AHP/BC mix rose to 327 °C. In Figure 2b, the thermal degradation of the BC/AHP system also exhibited two stages; the corresponding *T*_max_ were 330 °C and 431 °C, respectively, which was significantly lower than pure AHP. This may be attributed to the thermal degradation of AHP promoted by BC, leading to increased pyrolysis gasses such PH_3_, H_2_O, and so on [14], which may help isolate oxygen from the matrix and reduce the maximum thermal degradation rate of the composite. At 800 °C, the carbon residue rises to 81.1%, caused by the high heat generated from BC on its own. As it forms a uniform, dense carbonized layer, the BC/AHP system appears to effectively improve the flame retardancy of the PLA composites.

The TG and DTG curves of the different BC/PLA composites are shown in Figure 2c,d and corresponding data are given in Table 3. The trends are very similar to TG and DTG curves in previous work by Qian et al., (2016) [7] for PLA containing different BC levels up to 40%. The *T*_−5%_ of pure PLA was 331 °C and it almost completely decomposed at 600 °C, with a residue yield of 0.9%. The rapid thermal degradation is reflected in that the *R*_peak_ reached 27.5%/min at 365 °C. From Figure 2c, BC/PLA composites exhibited a single thermal degradation trend. From Table 2 and Table 3 the BC/PLA mix reduced the onset temperature of thermal degradation (*T*_−5%_) and increased the residue yield with greater levels of BC addition. The *T*_−5%_ of the BC/PLA (65/35) composite dropped to 234 °C and residue at 600 °C was 8.2 times higher than that of 5% BC addition, and 67 times that of pure PLA. *R*_peak_ decreased by 26.3% at 328 °C, which is only 4.5% lower than pure PLA, but the char residue yield at 600 °C increased from 0.9% to 24.9%. From the DTG curve (Figure 2b), the addition of BC significantly reduced the *R*_peak_ of BC/PLA composites. BC is a controlled pyrolysis product of bamboo with higher thermal stability than PLA. The thermal decomposition peaks decreased significantly with the co-addition of AHP. Figure 2e,f show the TG and DTG curves of PLA, BC (25%)/PLA, and BC (25%)/PLA/AHP composites under nitrogen atmosphere, and the corresponding data are given in Table 4. From Figure 2f, the 600 °C carbon residue yield also increased greatly, to 40.3% at 30% AHP co-addition, compared with adding BC only, an increase of 62%. The *T*_−5%_ with AHP addition was reduced by 11 °C from pure PLA, and BC/AHP addition reduced *T*_−5%_ by 4 °C. Pure PLA decomposes in a single step at 367 °C, leaving 0.3% char residue at 800 °C [10]. When AHP is added, the PLA composite starts decomposing earlier, but also in a single step with a shoulder at around 320 °C, with little influence on the *T*_max_ compared with pure PLA [10].

### 3.3. Elemental Analysis of Thermal Degradation Products

The FTIR spectra for PLA, BC/PLA, and BC/PLA/AHP mixes after CCT are shown in Figure 3a and the XPS spectra for the BC/PLA/AHP (25/50/25) composite are shown in Figure 3b. Figure 3c,d show the XPS P_2p_ spectra and the XRD patterns for crystallization in the BC/PLA/AHP after thermal degradation. The main signals are assigned and summarized in Table 5.

From Figure 3a, the absorptions at 2943 cm^−1^ are assigned to –CH_2_–, 1752 cm^−1^ to C=O, and 1456 cm^−1^ to C–H. The intensity was weakened with the addition of BC and BC/AHP, indicating that BC and BC/AHP fillers promoted the earlier degradation of PLA. This is consistent with the reduced thermal degradation temperatures of the composites. The absorption at 2359 cm^−1^ is assigned to P–H, which could be attributed to the generation of H_3_PO_4_ in condensed phase under pyrolysis. Moreover, 1070 cm^−1^ is assigned to PO_4_^3-^ formed during thermal degradation of AHP, and the peak intensity increased with higher levels of AHP addition. Further, 480 cm^−1^ is assigned to Al–O [17], which is present in the solid residues of composites containing AHP. Finally, 1182 cm^−1^ is assigned to PO_2_ [15] and 1070 cm^−1^ is assigned to P–O–P [16]. The new absorptions at 480 cm^−1^, 1070 cm^−1^, and 1182 cm^−1^ indicate that the BC/PLA/AHP composites generated Al_4_(P_2_O_7_)_3_ during thermal degradation [18].

From the XPS spectra (Figure 3b), C makes up the majority of the residue elemental balance, followed by O and P, whereas the Al fraction is low. In Figure 3c, the binding energies of P_2p_ in P–O–P, O=P–O^−^, and P–N–C are 134.7 eV, 133.2 eV, and 136.2 eV, respectively, indicating that stable condensed phase flame retarding substances were formed during thermal degradation of AHP [19]. Depending on the amount of AHP added, theoretically, the residue should contain 9.34 wt. % of P, whereas the actual P only accounted for 6.48 wt. %, indicating that some of the elemental P was converted to gas, namely PH_3_. AHP employs an effective gas phase flame retardant [20,21], whereby non-flammable gasses such as PH_3_ dilute and react with flammable gases, mainly O_2_, helping protect the internal matrix from further thermal degradation. The flame-retardant mechanism of AHP is discussed further later.

Crystallinity can be affected during combustion by the production of CO_2_, which, in higher concentrations, induces greater chain mobility PLA, which in turn accelerates the crystallization process [22]. In the XRD spectra, for different levels of AHP (Figure 3d), the small peak at 2θ = 15.9° and the larger peak at 26.6° are characteristic peaks of AHP [23]. The peak at 2θ = 20.9° belongs to the crystal plane of C (002), indicating the graphitized structure of BC. Its intensity decreased with increasing AHP addition, and the peak area also decreased, suggesting reduced graphite crystal formation. Pure PLA has an amorphous state, with no observable intensity peaks at 2θ = 15.9° or elsewhere on the XRD spectrum; strong peaks at 2θ = 15.9° become apparent when the PLA is foamed using a CO_2_ blowing agent, and particularly if the PLA is annealed at 100 °C [22]. This demonstrates that thermal degradation of the PLA composites here produced greater emissions of CO_2_.

### 3.4. TG-FTIR for Gas Phase Pyrolysis Products

To further examine the gas phase pyrolysis products of pure PLA and the composites with added BC and AHP, TG-FTIR was used to characterize specific products during pyrolysis under N_2_. Three-dimensional FTIR spectra of evolved gaseous products are shown in Figure 4, showing characteristic peaks around 3400–4000, 2700–3000, 2100–2400, 1700–1800, 1300–1450, and 1000–1250 cm^−1^. Reference FTIR peaks are H_2_O (3400–4000 cm^−1^), CO_2_ (2300–2400 cm^−1^), compounds containing carbonyl groups (1760 cm^−1^), and methyl-substituted compounds (3000 cm^−1^) [24,25,26].

Figure 5 shows selected FTIR spectra isolated from the 3D-FTIR for different temperatures ranging from 250 to 600 °C. The main gas phase products of PLA pyrolysis are H_2_O (3735 cm^−1^), hydrocarbons (1370 and 3000 cm^−1^), CO_2_ (2360 cm^−1^), CO (2175, 2108 cm^−1^), carbonyl group (1760 cm^−1^), aromatic compounds (1412 cm^−1^), and aliphatic compounds (1122 cm^−1^). From Figure 5a, for pure PLA, the peak of CO_2_ (2360 cm^−1^) appears at 250 °C, whereas CO (2175, 2108 cm^−1^) appears at 366 °C. At 600 °C, hydrocarbons (1370 and 3000 cm^−1^) containing carbonyl group (1760 cm^−1^), aromatic compounds (1412 cm^−1^), and aliphatic esters (1122 cm^−1^) start to weaken. These peaks reached their maximum values, reflecting the maximum weight loss rate of pure PLA at around 366 °C. From Figure 5b, the BC/PLA (25/75) mix exhibited new peaks at 1564 cm^−1^ and 1060 cm^−1^ corresponding to C=C and C–O, respectively. This suggests different gas-phase products promoted by further carbonization of the BC. The BC/PLA/AHP (25/50/25) shown in Figure 5c exhibits a new peak at 2320 cm^−1^, corresponding to P-H bonds, consistent with the formation of phosphine gas (PH_3_) during thermal decomposition of AHP, PLA, BC/PLA 25/75, and BC/PLA/AHP 25/50/25 composites at *T*_max_. This new peak also appears (denoted by red line) in the absorption spectra at *T*_max_ for BC/PLA/AHP (25/50/25), as shown in Figure 6.

Figure 7 shows the main pyrolysis products detected from the different selected materials as a function of temperature during TG-FTIR. Peak PH_3_ release (Figure 7a) occurred at a lower temperature in the BC and BC/AHP composites compared with neat PLA. Mixes without AHP showed one release event for PH_3_, whereas two peaks occurred when AHP was added. Similarly, Tang et al., (2013) [14] observed a large initial release of PH_3_ and a smaller second release event for PLA composites containing AHP. Peak hydrocarbons release (Figure 7b) occurred earlier at 350 °C for the BC composite compared with 400 °C for pure PLA and BC/AHP composite. Peak hydrocarbons were reduced in the BC/PLA/AHP composite. Interestingly, CO generation (Figure 7c), aromatic compounds (Figure 7e), and carbonyl groups (Figure 7f) were much lower in the BC/PLA composite compared with both pure PLA or the BC/PLA/AHP composite. Pyrolysis gases from pure PLA combustion are very rich in CO_2_, CO, H, and low molecular weight hydrocarbons such as methane, ethylene, ethane, propene, propane, and butanes [27]. PLA likely degraded differently in the presence of BC admixture without the mineral fire retardant (AHP) addition. The BC similarly absorbed heat earlier, and it is possible that some of the normal oxidation products from PLA thermal degradation reacted or were absorbed before being released in large quantities for detection. However, CO_2_ generation (Figure 7d) was higher, with an earlier peak for the BC/PLA than pure PLA. This is in accordance with TGA, showing the earlier onset of thermal degradation of the PLA when BC is added. Another mode of action influencing the pyrolysis gas emissions, particularly the aromatics and carbonyl groups from the BC/PLA composite, may be the very good gas absorption and adsorption properties of BC [28]. It is possible that the diffusion of certain thermal degradation gasses through the matrix could be retarded by the presence of absorbent BC particles.

During combustion of organic materials, CO and CO_2_ are primary products of the carbon–oxygen reaction. CO is generated first, which reacts with O_2_ to form CO_2_ [29]. A study by Du et al., (1991) [30] demonstrated how the presence of Ca in char particles strongly favours the formation of CO_2_ as the substrate combusts; at 390 °C, the Ca content in char particles reduced the CO/CO_2_ product ratio from 0.53 to 0.007. The co-addition of BC and AHP reduced overall CO_2_ production, but increased non-flammable gases (PH_3_, CO) during thermal degradation, forming a gas-phase flame-retardant barrier outside the material, thereby reducing the degradation rate of the internal material. The AHP addition appears to have significantly impeded the full combustion process for the material and its accompanying conversion of CO to CO_2_, as occurred in the plain PLA and the BC/PLA. The results are consistent with the TG analysis, showing that BC and co-added BC + AHP accelerate the thermal degradation of the PLA phase to form a carbonized layer, which impedes oxygen from accessing the inner layers of the composite. The flame-retardant mechanism is discussed further in the next section. Note from Figure 7 that, of all the gas compounds released over the course of temperature increase, only PH_3_ and CO_2_ had further increasing detection levels after their initial peak at around 350 to 420 °C. CO and PH_3_ are also the most toxic, and further research on toxic smoke generation from the BC/PLA/AHP prototype materials, especially if they are produced as thin shells or foams, is needed.

## 4. Discussion—Theoretical Model for Flame-Retardant Mechanism

As plastic polymers are subject to high temperatures, the polymer chains undergo scission reactions, releasing a complex range of combustible volatiles that self ignite at the flash point for the polymer via reactions with O_2_ to form highly reactive free radicals such as H· and OH· [31]. In other words, virgin PLA that has already been subjected to a high temperature environment, such as film blowing or the melt blending and extrusion/pressing processes here, is already predisposed to earlier thermal degradation at lower temperatures. At temperatures above about 300 °C (well above process temperature), pyrolysis becomes dominated by non-oxidative thermal decomposition [32].

AHP is an active inorganic (mineral) “filler” and fire retardant commonly used for protecting polymers, and is cost effective compared with intumescent flame retardants (IFRs) [33,34]. It is an active flame retardant that undergoes endothermic early degradation, absorbing energy and releasing non-flammable compounds such as PH_3_, H_2_O, and CO_2_, which dilute the combustible volatiles from polymer combustion as well as contribute to the formation of a protective ceramic or ‘vitreous’ layer [31]. Other contributory mechanisms of mineral fillers, even if inert, include reduced combustible materials per weight, modified thermal conductivity of the composite and its thermophysical properties, and melt viscosity of the resulting material [31]. The BC could be considered to be an inert, mostly carbonized filler that contributes to the flame-retardant composite in the abovementioned ways, including reduction of the quantity of flammable PLA present in the composite.

On the basis on the thermal behavior and elemental analyses presented, a hypothetical flame-retardant mechanism for BC/PLA/AHP system is summarized schematically in Figure 8.

As heating progresses, the presence of BC powder in the PLA caused the early onset of thermal degradation (as shown in Figure 2a,c) and char formation in the PLA, which forms an effective condensed flame-retardant carbonized layer over the surface of the composite. The thermal conductivity (TC) of the composite, while not measured directly here, is likely significantly higher than pure PLA, and the TC of the carbonized bamboo filler is greater than the PLA. TC of PLA-based composites is linearly positively correlated with material density, with pure PLA having low TC for its density and glass fibre-reinforced plastics having high TC relative to density (Tagaki et al., 2002) [35]. The TC of pure PLA is low (between 0.13 and 0.2 W/(m·K)), but very effectively enhanced by the addition of fine carboniferous materials like graphene (Lin et al., 2018) [36]. Chiang et al., (2013) [37] recorded an increase in the TC of epoxy resin from 0.17 W/(m·K) to 0.23 with the addition of 2% BC powder, and a study by Tagaki et al., (2002) [35] showed how adding 60% *w*/*w* bamboo fibre to PLA increased its solid phase TC from 0.2 W/(m·K) to 0.34 W/(m·K), with modelling indicating the TC of solid bamboo fibre substrate is greater than 0.34 W/(m·K). The TC of pure phase highly porous chars like BC is difficult to measure experimentally, and requires theoretical estimates (Staggs 2002) [38]. TC of porous carbonized materials decreases with the increasing porosity and increases with temperature (Staggs 2002) [38]. Verma et al., (2019) [39] measured the thermal conductivity of activated carbon sheets bonded with epoxy for cryocooling applications at between 0.3 and 0.63 W/(m·K), for temperatures between 4.5 K (−268.65 °C) and 300 K (27.85 °C), considerably higher than PLA.

The flame-retardant mechanism of AHP itself is a dual mode of very effective gas phase and condensed phase [2,40]. It relies on early onset thermal degradation and recombination reactions after absorbing heat energy to form a gas phase flame-retardant phenomenon, which hampers further heat and oxygen transfer through the composite and helps to increase the carbon residue rate of the BC/PLA/AHP composite. During combustion, the AHP decomposes in two stages [14,41,42]; first, at 322 °C to form phosphine (PH_3_) and aluminum hydrogen phosphate [Al_2_(HPO_4_)_3_]:2Al_2_(H_2_PO_2_)_3_→Al_2_(HPO_4_)_3_ + 3PH_3_(2)
and second at 430 °C, whereby the Al_2_(HPO_4_)_3_ degrades to aluminum pyrophosphate [Al_4_(P_2_O_7_)_3_], releasing water:2Al_2_(H_2_PO_2_)_3_→Al_4_(P_2_O_7_)_3_ + 3H_2_O(3)

PH_3_ oxidizes to form phosphoric acid (H_3_PO_4_) and phosporous pentoxide (P_2_O_5_), consuming O_2_ and releasing H_2_O, which helps retard further combustion of the material [43]. The AHP addition appears to have significantly retarded the conversion of CO to CO_2_, as occurred in the pure PLA and the BC/PLA mixes. The O_2_ scavenging effect of the earlier release of PH_3_ in the BC/AHP composite hampered the full combustion cycle and conversion of CO to CO_2_, as observed from the TG-FTIR pyrolysis products. H_3_PO_4_ reacts to form phosphite ions (PO_2_^3−^), and phosphate ions (PO_4_^3−^), which are able to quench reactive free radicals including H· and OH·. Phosphorous from thermally degraded AHP is also converted to a condensed phase producing a continuous and dense inorganic carbonized layer containing Al_4_(P_2_O_7_)_3_, which helps block heat and oxygen transfer through the surface, protecting the internal matrix [43]. There must be sufficient ingredient to form a continuous, dense, non-porous char layer to provide effective protection against further combustion [31]. In the case of AHP, it is relatively high—30% to 40% by weight, referring to the results of flame-retardant tests.

## 5. Conclusions

A new combination of BC and AHP was investigated for improving the flame retardancy of PLA matrix composites. The BC and AHP fillers promote the early heterogeneous nucleation of PLA, causing it to crystallize and become brittle at lower temperatures. AHP and BC fillers not only reduce the quantity of flammable PLA, but also provide a dual action flame retardant: a condensed phase (protective surface char formation) and a gas phase (oxygen dilution and consumption). The condensed phase is the early onset thermal degradation of the surface material, forming a dense layer of char over the surface, which impedes further combustion of inner material. The co-addition of BC and AHP reduced the onset temperature, *T*_−5%_, for thermal degradation of PLA and increased the carbonization rate at 600 °C. Thirty percent (30%) co-addition of AHP significantly reduced the thermal decomposition peaks of the BC/PLA composites and increased the protective carbon residue or ‘char’ production rate to 40.3% or 43 times that of pure PLA. The gas-phase involves the generation of non-combustible gases such as PH_3_, which reacts with and consumes O_2_ to produce H_2_O vapour. Other non-flammable combustion gasses include CO, which reacts with O_2_ to form CO_2_, further consuming and diluting oxygen at the surface and helping retard oxygen flow to the subsurface. The BC/PLA composite produced the most CO_2_, but release of CO, aromatic compounds, and carbonyl groups was heavily suppressed. Adding AHP greatly reduced CO_2_ emission, which was thought to be from a combination of O_2_ scavenging from the earlier release of reactive PH_3_ gas, and incomplete pyrolysis of the matrix. BC addition on its own produced a more thermally conductive composite, accelerating degradation of the surface layers to char, but its ability to contribute a gas phase flame retardancy effect, like AHP or other active flame-retardant mineral fillers, is unclear. Further work will focus on fabricating and testing a foamed version of the BC/PLA/AHP composite system for key properties and fire retardancy.

## Figures and Tables

**Figure 1 polymers-12-02238-f001:**
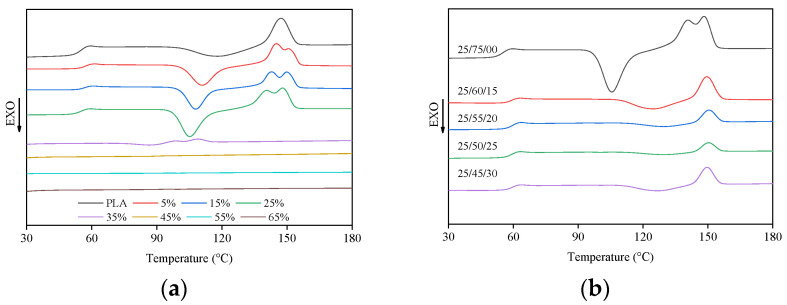
Heat flow curves of (**a**) PLA and BC/PLA, (**b**) BC/PLA/AHP with different amount of AHP added.

**Figure 2 polymers-12-02238-f002:**
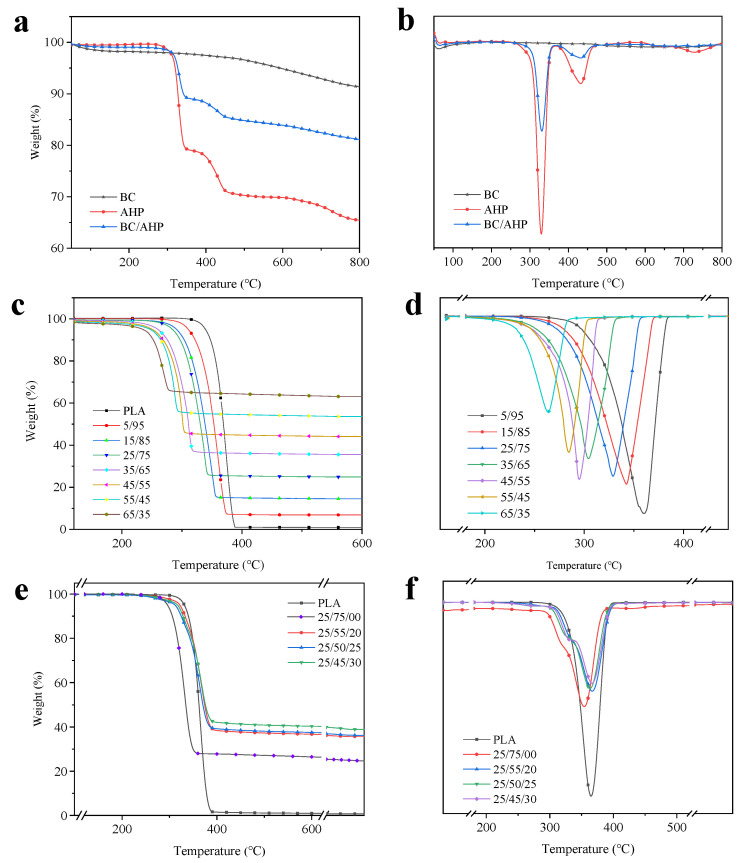
TG and DTG curves for (**a**,**b**) BC and AHP, (**c**,**d**) BC/PLA mixes, and (**e**,**f**) BC/PLA/AHP mixes.

**Figure 3 polymers-12-02238-f003:**
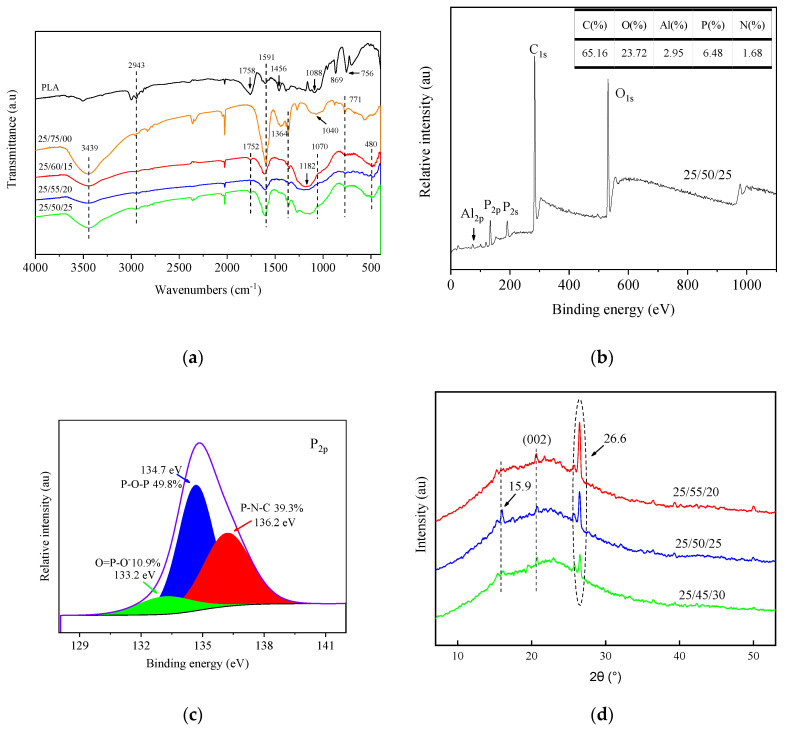
(**a**) Fourier transform infrared (FT-IR) spectra of PLA, BC/PLA, and BC/PLA/AHP; (**b**) X-ray photoelectron analysis (XPS) spectra of BC/PLA/AHP (25/50/25); (**c**) XPS P_2p_ spectra of BC/PLA/AHP (25/50/25); and (**d**) X-ray diffraction (XRD) patterns of BC/PLA/AHP after thermal decomposition.

**Figure 4 polymers-12-02238-f004:**
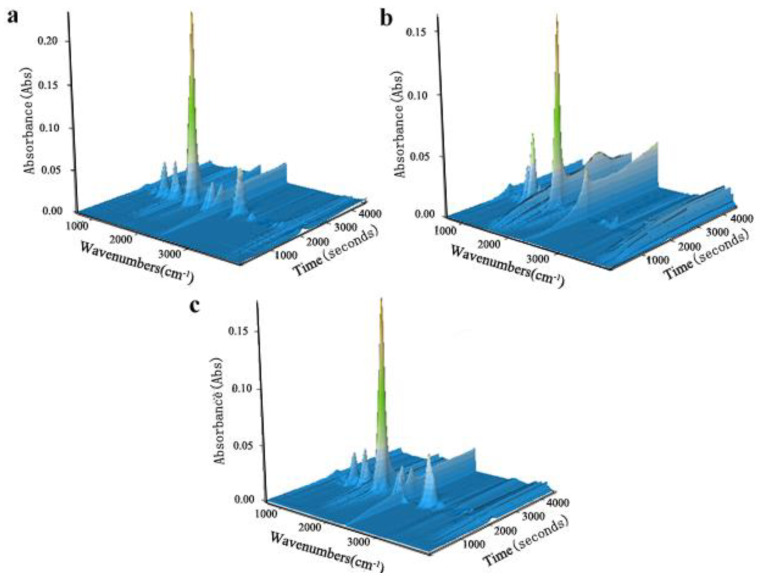
Three-dimensional thermogravimetry-Fourier transform infrared spectrometry (TG-FTIR) spectra of gaseous products in the thermal decomposition of PLA and flame-retardant composites: (**a**) PLA, (**b**) BC/PLA, and (**c**) BC/PLA/AHP.

**Figure 5 polymers-12-02238-f005:**
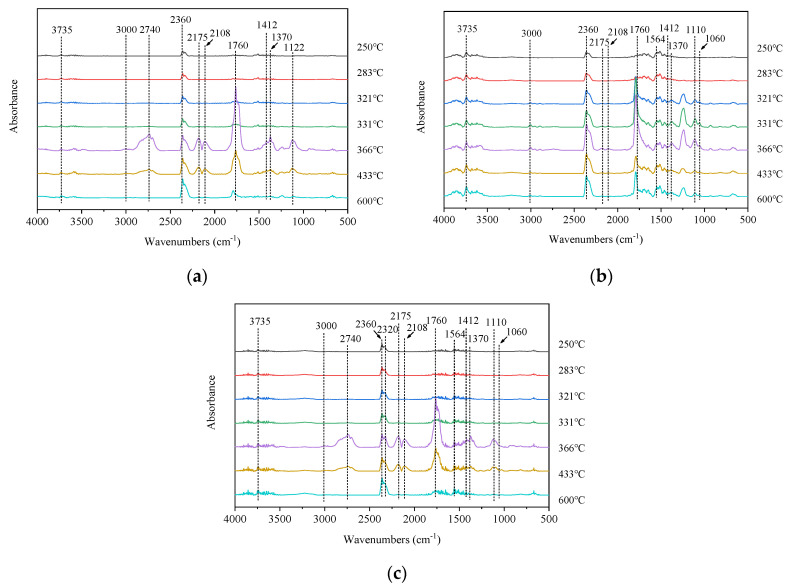
FTIR spectra of gas phase products of (**a**) PLA, (**b**) BC/PLA, and (**c**) BC/PLA/AHP at various temperatures.

**Figure 6 polymers-12-02238-f006:**
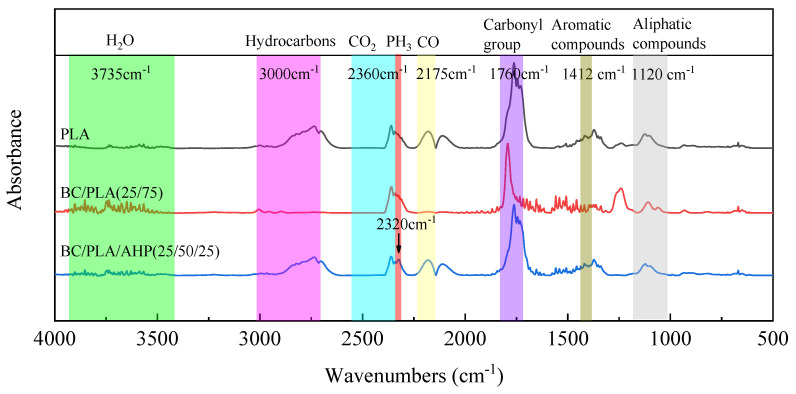
FTIR spectra of gaseous products for the thermal decomposition of PLA, BC/PLA 25/75, and BC/PLA/AHP 25/50/25 composites at *T*_max_.

**Figure 7 polymers-12-02238-f007:**
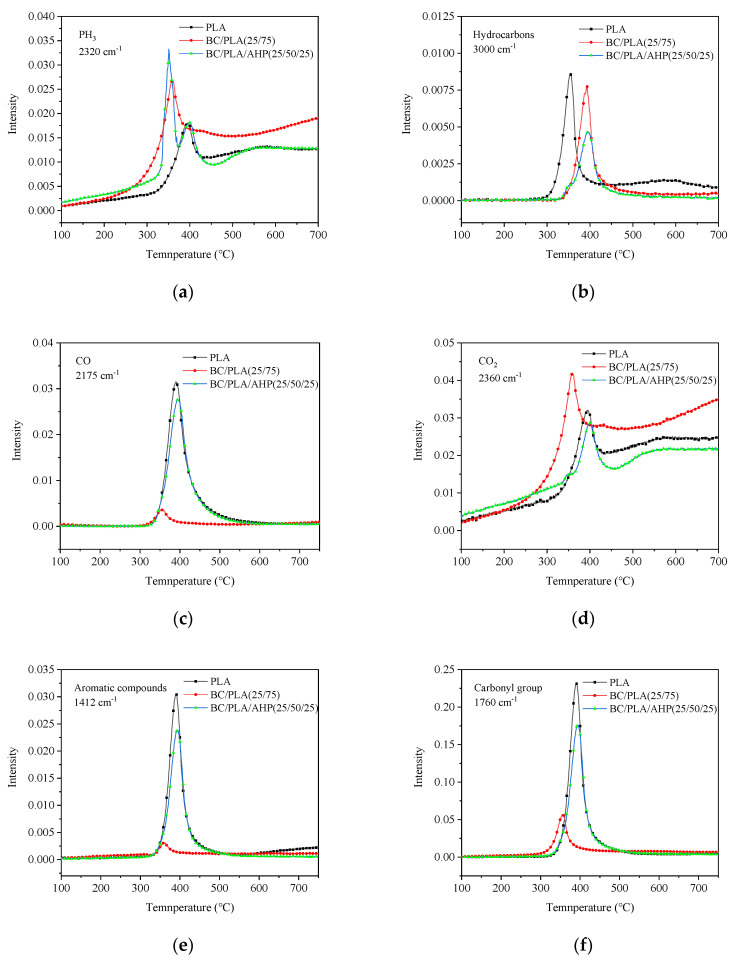
Absorbance spectra for pyrolysis products (**a**) PH_3_, (**b**) hydrocarbons, (**c**) CO, (**d**) CO_2_, (**e**) aromatic compounds, and (**f**) carbonyl groups for PLA, BC/PLA, and BC/AHP at various temperatures.

**Figure 8 polymers-12-02238-f008:**
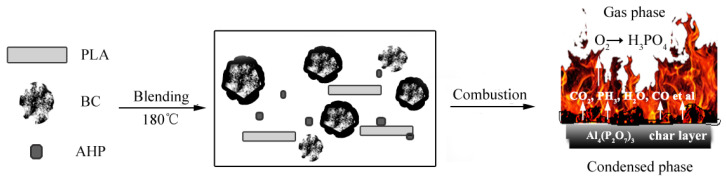
Schematic representation of the flame-retardant mechanism of BC/PLA/AHP during combustion.

**Table 1 polymers-12-02238-t001:** DSC testing data of PLA, BC/PLA, and BC/PLA/AHP.

Sample	*T*_g_(°C)	*T*_cc_(°C)	Δ*H*_c_(J/g)	*X*_c_(%)	*T*_m1_(°C)	*T*_m2_(°C)	Δ*H*_m_(J/g)
PLA	54.59	118.00	10.51	11.23	147.13	-	13.45
BC/PLA/AHP (5/95/00)	57.28	111.35	21.18	23.82	145.19	151.96	20.09
BC/PLA/AHP (15/85/00)	56.89	107.93	19.89	25.00	142.89	150.77	24.55
BC/PLA/AHP (25/75/00)	55.13	105.54	19.59	27.91	139.95	148.16	21.41
BC/PLA/AHP (35/65/00)	-	86.72	1.58	2.60	97.83	108.76	1.42
BC/PLA/AHP (25/60/15)	59.40	124.92	10.63	17.81	149.55	-	10.24
BC/PLA/AHP (25/55/20)	59.28	129.28	3.22	5.73	150.62	-	4.48
BC/PLA/AHP (25/50/25)	58.90	129.17	2.14	4.07	150.63	-	3.33
BC/PLA/AHP (25/50/30)	59.21	127.02	6.50	13.23	-	149.36	6.36

**Table 2 polymers-12-02238-t002:** Thermogravimetry (TG) test data of BC, AHP, and BC/AHP flame-retardant systems under N_2_ atmosphere.

Sample	*T*_−5%_ (°C)	*R*_1peak_/*T*_1max_ (%·min^−1^/°C)	*R*_2peak_/*T*_2max_ (%·min^−1^/°C)	Carbon Residue Rate/%
				800 (°C)
BC	-	0.2/649	-	91.4
AHP	320	7.5/331	1.6/433	65.5
BC/AHP	327	3.5/330	0.6/431	81.1

**Table 3 polymers-12-02238-t003:** TG test data of PLA and BC/PLA materials under N_2_ atmosphere.

Sample	*T*_−5%_ (°C)	*R*_peak_/*T*_max_ (%·min^−1^/°C)	Carbon Residue Rate/%		
			400 °C	500 °C	600 °C
PLA	331	27.5/365	1.5	1.1	0.9
BC/PLA (5/95)	310	32.4/359	7.1	6.9	6.9
BC/PLA (15/85)	291	27.4/341	15.0	14.7	14.5
BC/PLA (25/75)	283	26.3/328	25.4	25.1	24.9
BC/PLA (35/65)	259	23.4/304	36.2	35.8	35.6
BC/PLA (45/55)	250	26.8/295	44.8	44.3	44.1
BC/PLA (55/45)	247	22.4/284	54.5	54.0	53.6
BC/PLA (65/35)	234	15.7/267	64.3	63.6	63.1

**Table 4 polymers-12-02238-t004:** TG test data of PLA, BC/PLA, and BC/PLA/AHP materials under N_2_ atmosphere.

Sample	*T*_−5%_ (°C)	*R*_peak_/*T*_max_ (%·min^−1^/°C)	Carbon Residue Rate/%		
			400 °C	500 °C	600 °C
PLA	331	27.5/365	1.5	1.1	0.9
BC/PLA (25/75)	283	26.3/328	25.4	25.1	24.9
BC/PLA/AHP (25/55/20)	312	12.3/363	39.2	38.1	37.5
BC/PLA/AHP (25/50/25)	321	12.6/366	38.4	37.3	36.7
BC/PLA/AHP (25/45/30)	317	11.5/366	42.1	40.9	40.3

**Table 5 polymers-12-02238-t005:** Characteristic attribution of Fourier transform infrared (FTIR) absorption bands.

Wavenumber (cm^−1^)	Assignment
2943	Asymmetric stretching of CH_2_
2359	Stretching vibration of the P-H bond
1752	Out-of-plane stretching of C=O
1456	Stretching vibration of the C–H bond
1182	Bending vibration of PO_2_ [15]
1070	Bending vibration of P–O–P bond [16]
480	Stretch modes of Al–O bond

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
