# Peer review of "Development of Biodegradable Flame-Retardant Bamboo Charcoal Composites, Part II: Thermal Degradation, Gas Phase, and Elemental Analyses"

_polymers, 2020, doi:10.3390/polym12102238_

Round 1

Reviewer 1 Report

Dear Authors,

In my opinion, the part dedicated to the explanation of the thermal degradation and flame retardant mechanism is the most important for studies dedicated to the development of FR composites. Therefore, this part 2 makes it sense. Unfortunately, this article cannot be published as well. It is necessary to revise it deeply.

It is mandatory to completely rewrite the introduction. In its current state, it does not respond to the instructions given by "Polymers" (instructions that are always visible on line 41-48 !). The structuring of the introduction needs to be reviewed. It is quite surprising that the introduction ends with "Using BC may be more efficient". The lines 66 to 68 seem more appropriate to conclude an introduction.

Some terms are introduced in part 1. But it is necessary to repeat it again in the part 1. Readers could only study part 2 without reading part 1. For example, explain what means CCT.

The analyses (3.1) dedicated to crystallization properties (it lacks a "a" in the title of this part bring little benefit to the understanding of the thermal and flame retardant behaviour which are the topics of this article. They should be more adequate to discuss about mechanical behaviour. By the way, you recall a conclusion on line 155-156.

Some conclusions made in the paper should be supported by the analyses.

ex:

line 194 "leading to increased pyrolysis gasses such PH3, ..." please add a direct link with FTIR experiments

line 378:" BC absorbs heat faster"  on what analysis this conclusion is based? please help the reader!

==> what is the thermal conductivity of the BC?

This article is sometimes confusing. The authors should clearly indicate in the text as well as the legend of the figures the nature of the studied sample. For example, on which sample is made the XPS analyses presented in Figure 3b? In line 316-317, it is mentioned that Figure 7 shows the main pyrolysis products detected during cone calorimeter experiments. There is no mention of such experiments in part 2 "Materials and Methods". Furthermore, there are some mistakes in the text: line 269 and 265, FTIR bands are not well assigned (it doesn't correspond with the table 5. Please also add a reference for table 5).

To finish, the paragraph for XRD experiments is not clear. Why do you make a link with crystallinity and mechanical properties (line 278)? If I well understood, it is XRD data of residues. So what is the link?

Reviewer 2 Report

The manuscript titled: Development of biodegradable flame-retardant bamboo charcoal composites, Part II: Thermal degradation, gas phase and elemental analyses" by Wang et al. is well written, sufficiently referenced and generally does not oversell its main findings on the synergistic effect of BC and AHP. BC and AHP have been used individually as they have referenced so the novelty has to be highlighted more. Although well written, this work seems incremental rather than novel.
